# p53 dynamics vary between tissues and are linked with radiation sensitivity

Jacob Stewart-Ornstein [1,5], Yoshiko Iwamoto[2], Miles A. Miller [2], Mark A. Prytyskach[2], Stephane Ferretti[3], Philipp Holzer[3], Joerg Kallen[3], Pascal Furet[3], Ashwini Jambhekar[1], William C. Forrester [4], Ralph Weissleder[1,2 ✉] & Galit Lahav [1✉]

Radiation sensitivity varies greatly between tissues. The transcription factor p53 mediates the response to radiation; however, the abundance of p53 protein does not correlate well with the extent of radiosensitivity across tissues. Given recent studies showing that the temporal dynamics of p53 influence the fate of cultured cells in response to irradiation, we set out to determine the dynamic behavior of p53 and its impact on radiation sensitivity in vivo. We find that radiosensitive tissues show prolonged p53 signaling after radiation, while more resistant tissues show transient p53 activation. Sustaining p53 using a small molecule (NMI801) that inhibits Mdm2, a negative regulator of p53, reduced viability in cell culture and suppressed tumor growth. Our work proposes a mechanism for the control of radiation sensitivity and suggests tools to alter the dynamics of p53 to enhance tumor clearance. Similar approaches can be used to enhance killing of cancer cells or reduce toxicity in normal tissues following genotoxic therapies.

[1] Department of Systems Biology and the Ludwig Center at Harvard, Blavatnik Institute at Harvard Medical School, Boston, MA, USA. [2] Center for Systems Biology, Massachusetts General Hospital, Boston, MA, USA. [3] Novartis Institutes for Biomedical Research, Basel, Switzerland. [4] Chemical Biology and Therapeutics, Novartis Institutes for Biomedical Research, Cambridge, MA, USA. [5] Present address: Department of Computational and Systems Biology, University of Pittsburgh Medical School, Pittsburgh, PA, USA. One-sentence summary: The temporal dynamics of p53 in vivo are connected with radiation sensitivity and tumor clearance. ✉email: ralph_weissleder@hms.harvard.edu; galit@hms.harvard.edu

The tumor-suppressing transcription factor p53 is a master regulator of stress responses. It is expressed throughout the body[1] and has a major role in controlling cell fate in response to stress[2]. Studies comparing wild-type and p53-deficient mice showed that p53 is necessary for apoptosis in response to radiation[3–6] and plays a key role in specifying the diverse radio-sensitivities of different tissues in the body[3–9]. While it is clear that p53 and its target genes are strongly induced by DNA damage and drive the response to irradiation[2,10,11], tissues with notably different sensitivity to radiation, such as the gut and lymphoid organs, have similar levels of p53[8,11], suggesting more subtle alterations to its function between tissues may drive radiosensitivity.

Recent studies in cell culture revealed that the temporal dynamics of p53 play an important role in controlling the fate of single cells[12–14]. Specifically, in response to ionizing radiation, p53 protein shows a series of oscillations in each cell resulting from the negative feedback loop with p53's target gene and inhibitor, the E3-ligase Mdm2[15]. Transient and oscillatory activation of p53 was found to be compatible with DNA repair and proliferation, while sustained p53 levels, obtained by chemical manipulation of the p53-Mdm2 feedback loop using an Mdm2 inhibitor, led to terminal fates such as apoptosis and senescence[12]. Note that Mdm2 inhibitors are in a class of molecules currently being evaluated in clinical trials for tumor therapy[16–18]. p53 oscillations post-irradiation have been observed in multiple cultured cells from human and mice[15,19], as well as in vivo using a p53 luciferase reporter mouse[20]. Knowing that p53 protein accumulates in normal tissues following radiation, we aimed to determine whether p53 dynamics would also be relevant to the p53-mediated responses to irradiation in vivo and, critically, if we could perturb these dynamics to alter the sensitivity of tumors or tissues to radiation.

Here we focus on gastrointestinal and lymphoid tissues, which are known to have different radiation sensitivities despite comparable levels of p53[7]. Using immunostaining of p53 after radiation, we find different patterns of p53 dynamics across the tested tissues. As was previously reported, both tissue types show induction of p53 protein immediately following radiation[21]; however, in the small and large intestines p53 levels drop after peaking at 2–3 h whereas in the spleen and thymus p53 levels and activity are sustained. We test the effect of an MDM2 inhibitor (Mdm2i) on p53 dynamics and show that a combination of radiation and MDM2i sustains p53 dynamics in the gastrointestinal system. Lastly, we extend these findings to a tumor model; we show that the MDM2 inhibitor in combination with radiation extends p53 signaling and improves tumor control. This study proposes a mechanism for the control of radiation sensitivity in tissues and highlights the importance of examining the dynamics of key factors in vivo to better understand tissue-specific behaviors.

## Results

To study the p53 response in vivo, we examined p53 levels in various murine tissues before and after ionizing radiation (IR) using immunofluorescence staining. We focused on two highly radio-responsive lymphoid organs (spleen and thymus) and two less sensitive tissues (small and large intestines), which have been shown to express comparable levels of p53[7]. We observed minimal p53 staining in the tissues of untreated mice and substantial staining across the four tested tissues after irradiation (Fig. 1a). Note that, as was shown in earlier work[8,10,21], p53 staining after radiation was not ubiquitous within tissues. This pattern was particularly clear in the small (jejunum) and large intestine (colon), which showed strong p53 staining of crypt cells, but not

the enterocytes (Fig. 1a). The position of these p53-positive cells in the crypt suggests their stem cell nature, consistent with their positive staining for the stem cell marker OLMF4 (Supplementary Fig. 1). Staining for γ-H2AX, a marker of DNA double-strand breaks, showed induction of DNA damage in all tissues (Fig. 1b). Interestingly, there were subtle inter tissue differences in γ-H2AX; in the small intestine, γ-H2AX was lowest in p53-positive cells at the crypt, and highest in p53-negative cells of the villi (Fig. 1c, d). To investigate whether proliferative state contributes to the anti-correlation in γ-H2AX and p53 levels in the small intestine, we co-stained for p53 and Ki67, a marker for cycling cells. This analysis revealed a substantial overlap between p53-positive and Ki67-positive cells in crypts (Fig. 1e–g), suggesting that p53 is preferentially induced in cycling cells. It has been previously suggested that the replicative and stem nature of the cell impact the kinetics of DNA repair and the choice of repair mechanism (for example, an emphasis on non-homologous end joining in stem cells), which can explain the relatively lower levels of γ-H2AX in cycling crypt cells[22,23].

To determine how p53 induction affected cellular outcomes in each tissue, we used TUNEL and cleaved Caspase-3 staining to monitor for signs of cell death. As expected, the radiosensitive tissues showed extensive TUNEL and cleaved Caspase-3 signal (Fig. 1h, Supplementary Figs. 2, 3). Specifically, the spleen and thymus exhibited more TUNEL-positive and cleaved Caspase-3-positive cells compared to the intestines, in which apoptosis was detected only among the p53-positive crypt cells (Fig. 1h, Supplementary Fig. 2). These results confirm previous studies arguing that p53 is induced in response to irradiation across tissues, with spatial correlations between p53 induction and apoptosis in the intestines[8,24–26]. Note that in the small intestine, even in the p53-positive crypts, cell death was relatively rare suggesting the existence of additional mechanisms regulating cell death beyond absolute p53 levels.

Recent studies in cultured cells have shown that in addition to the absolute levels of p53, the temporal dynamics of p53 play a role in the response to DNA damage[12–14]. To determine whether p53 exhibited dynamical changes in vivo, we quantified p53 levels over time in tissues using semi-automated image analysis. We found that in the small and large intestines, p53 levels increased 2 h after irradiation, and then declined close to background levels by 5 h (Fig. 2a, b, Supplementary Fig. 4). In contrast, spleen and thymus showed induction of p53 that remained elevated throughout the time course (Fig. 2a, b, Supplementary Fig. 4). Phospho-p53, an activated form generated by the DNA damage-responsive kinases such as ATM[27], exhibited similar tissue-specific dynamics (Fig. 2c). We next further examined p53 levels in the small intestine over time and its spatial position with respect to the bottom of the crypt. We found that p53 induction in the small intestine is rapid, transient, and highest in basal crypt cells (Fig. 2d, e). In contrast to the small intestine, the spleen showed a relatively uniform and sustained induction of p53 levels that peaked at 2 h and remained elevated for at least 7 h (Fig. 2f). To determine whether the dynamical patterns in p53 levels were reflected in the expression of its target genes, we quantified the mRNA expression dynamics of three canonical p53 target genes: (i) Mdm2, an E3 ubiquitin ligase that negatively regulates p53 stability; and whose expression tracks with p53 levels in cultured cells; (ii) CDKN1A (p21), a CDK inhibitor associated with cell cycle arrest; and (iii) PUMA, a BH3-only pro-apoptotic factor (Fig. 2g). The lymphoid organs (spleen and thymus) showed sustained transcription of all three tested genes, whereas the small and large intestines showed a transient rise in transcription, which resembled the previously described p53 oscillations in whole mouse experiments using luciferase[20] and in cultured cells[12,15,19]. Taken together, our results show that the

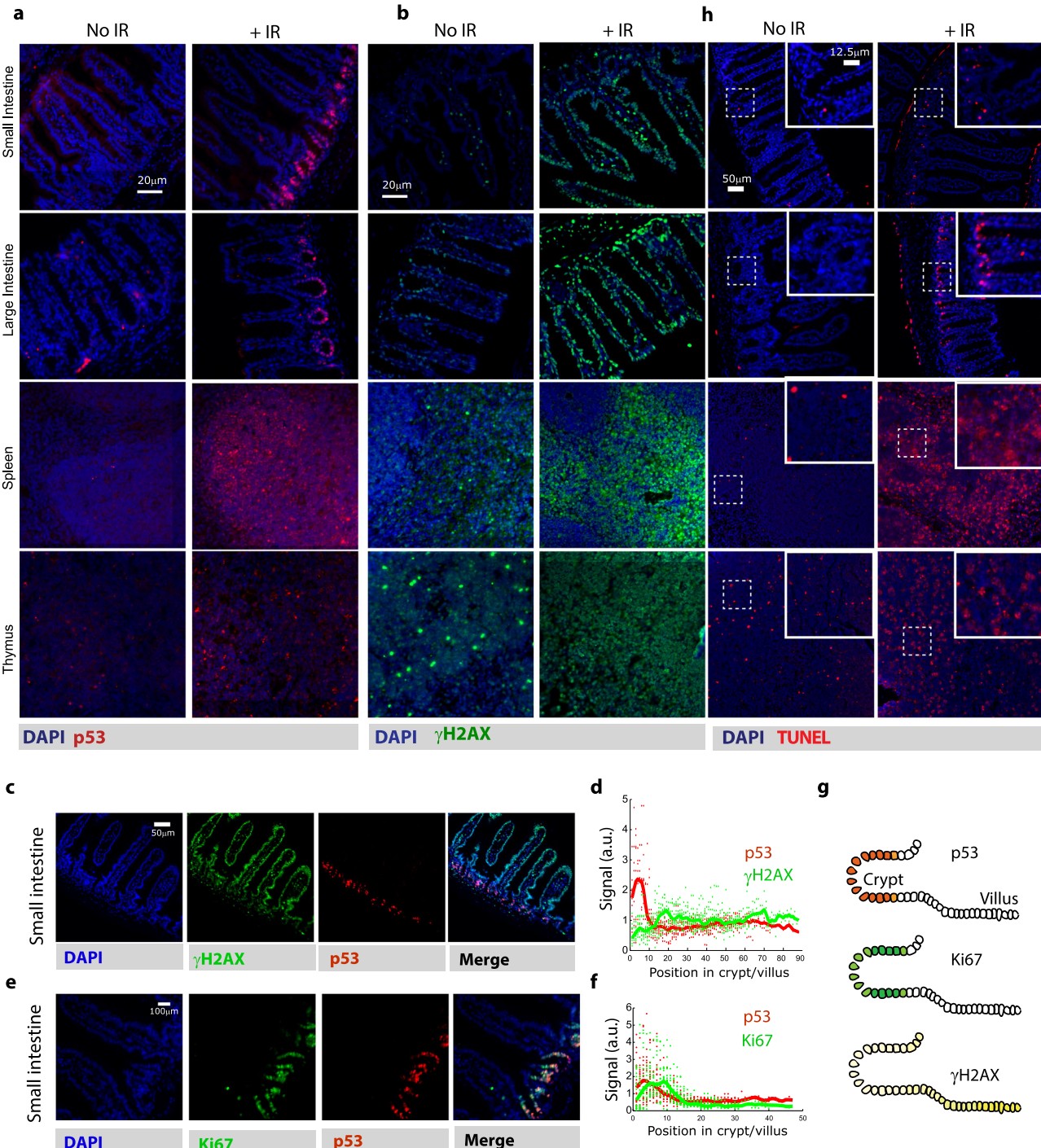

**Fig. 1 p53 is induced in irradiated tissues.** Mice were subjected or not to total body irradiation (IR) and after 2 h the indicated tissues were analyzed by **a** p53 immunofluorescence, **b** γ-H2AX immunofluorescence. Representative images from three (**a**) or two (**b**) independent experiments are shown. **c**, **e** Co-staining of p53 and γ-H2AX (**c**) or Ki67 (**e**) in the small intestine 2 h after irradiation. **d**, **f** Data from **c** and **e**, respectively, plotted as a function of cell position from the crypt base. Bold line shows running average; dots represent individual cells ($n = 461$ cells (**d**) and 615 cells (**f**)). **g** A diagram of crypt structure and summary of staining results in the small intestine. **h** TUNEL staining for indicted tissues in untreated mice or 5 h after ionizing radiation. Experiments were performed in duplicates.

dynamics of the p53 response varies across tissues: lymphoid organs that are known to be more prone to apoptosis after irradiation showed a sustained expression of p53 and its target genes, while the more resistant tissues, e.g., small and large intestines, showed a transient response.

We next asked whether the relatively lower radio-sensitivity of intestinal cells compared to lymphocytes[8] may result from the transient nature of p53 accumulation in the intestinal crypt cells (Fig. 2a–e). To alter p53 dynamics we used a novel Mdm2 inhibitor (Mdm2i), NMI801 (**N**ovartis **M**DM2 NVP-CD**I801**) (Fig. 3a). This is a fourth-generation potent and selective Mdm2 inhibitor that binds specifically to Mdm2 through interactions with its p53 binding pocket (Fig. 3b). NMI801 was designed based on an X-ray structure of p53 bound by Mdm2 and is derived from

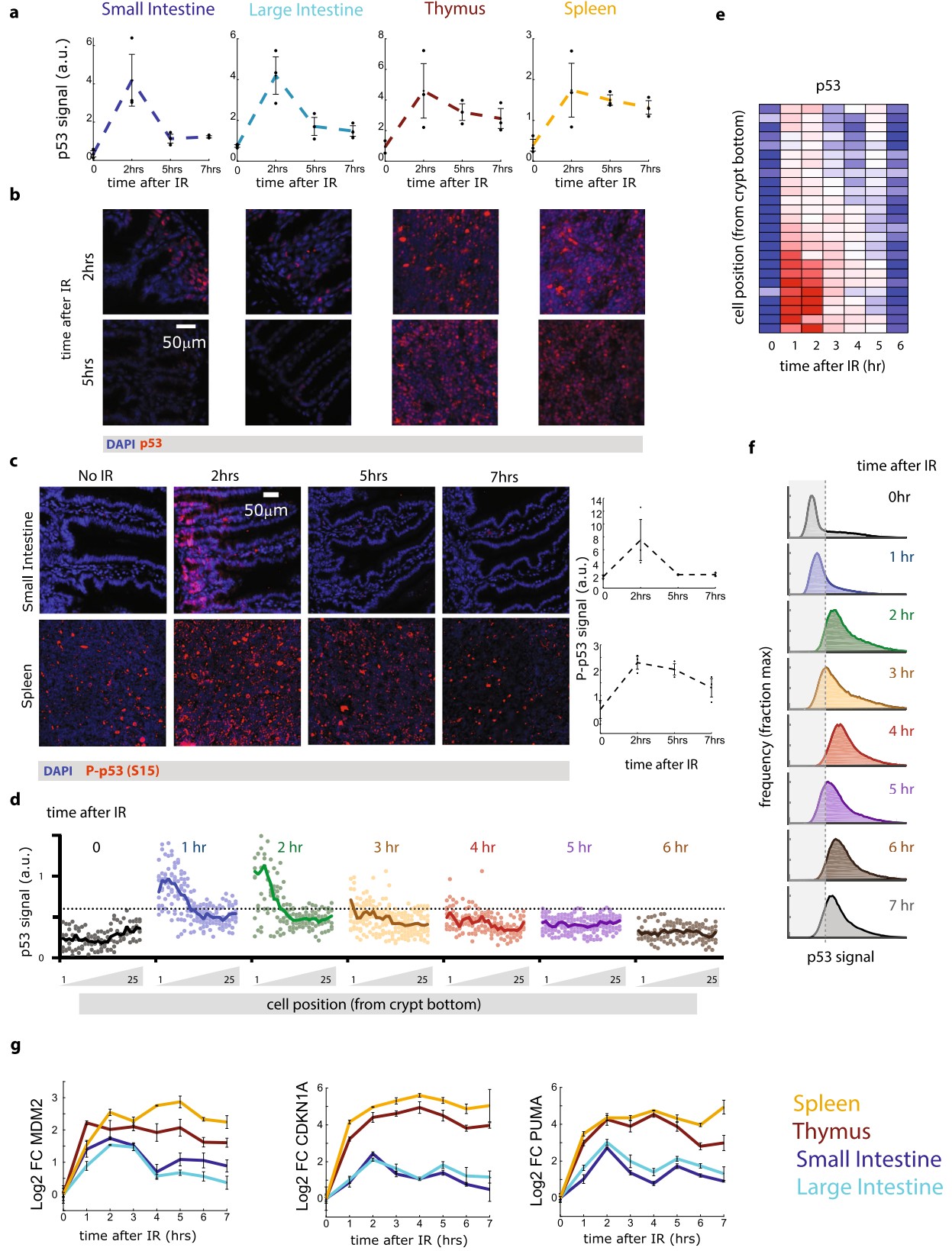

an indolyl-imidazole backbone[28,29]. NMI801 inhibited Mdm2 with a inhibitory concentration of ~1 nM across species, including human and mouse, as measured in vitro based on the ability of NMI801 to disrupt MDM2-p53 binding (Fig. 3c). Addition of NMI801 to cells in culture with wild-type p53 led to increased abundance of p53 in a dose-dependent manner (Fig. 3d). The

amplitude and duration of p53 levels increased with NMI801 dose, with the highest concentration (1 μM) leading to elevated p53 for approximately 8 h (Fig. 3e). To assess the effects of NMI801 in vivo, we treated mice bearing xenografts of the p53 wild-type colon cancer line HCT116 orally with 200 mg/kg per day for 10 days. This treatment suppressed tumor growth

**Fig. 2 p53 dynamics vary across tissues. a** Average p53 immunofluorescence intensity in the indicated tissues as a function of time after irradiation (IR, $n = 3$ samples per tissue, except for the thymus, in which $n = 2$ at $t = 0$; error bars are SEM. Dots indicate individual samples). **b** Representative images from three independent experiments of p53 immunofluorescence at 2 h or 5 h after irradiation. **c** Representative images and quantification of p53-phosphoS15 immunofluorescence in the small intestine and spleen as a function of time ($n = 3$ samples per tissue; Error bars are SEM. Dots indicate individual samples). **d** Quantification of p53 intensity across the crypt of the small intestine in mice treated with IR and analyzed at the indicated time points ($n = 125,225,175,275,200,225$ cells per time point). **e** Heat map (blue = low, red = high) of the average p53 intensity from **a** as a function of time and cellular position within the crypt. **f** Histograms of p53 immunofluorescence intensity in spleen cells at the indicated time points following irradiation (distributions calculated from >200 cells per time point). **g** qPCR-based quantification of the fold change (FC) in mRNA levels of p53 target genes MDM2, CDKN1A (p21), and PUMA in the indicated tissues as a function of time. Experiments were done in duplicates.

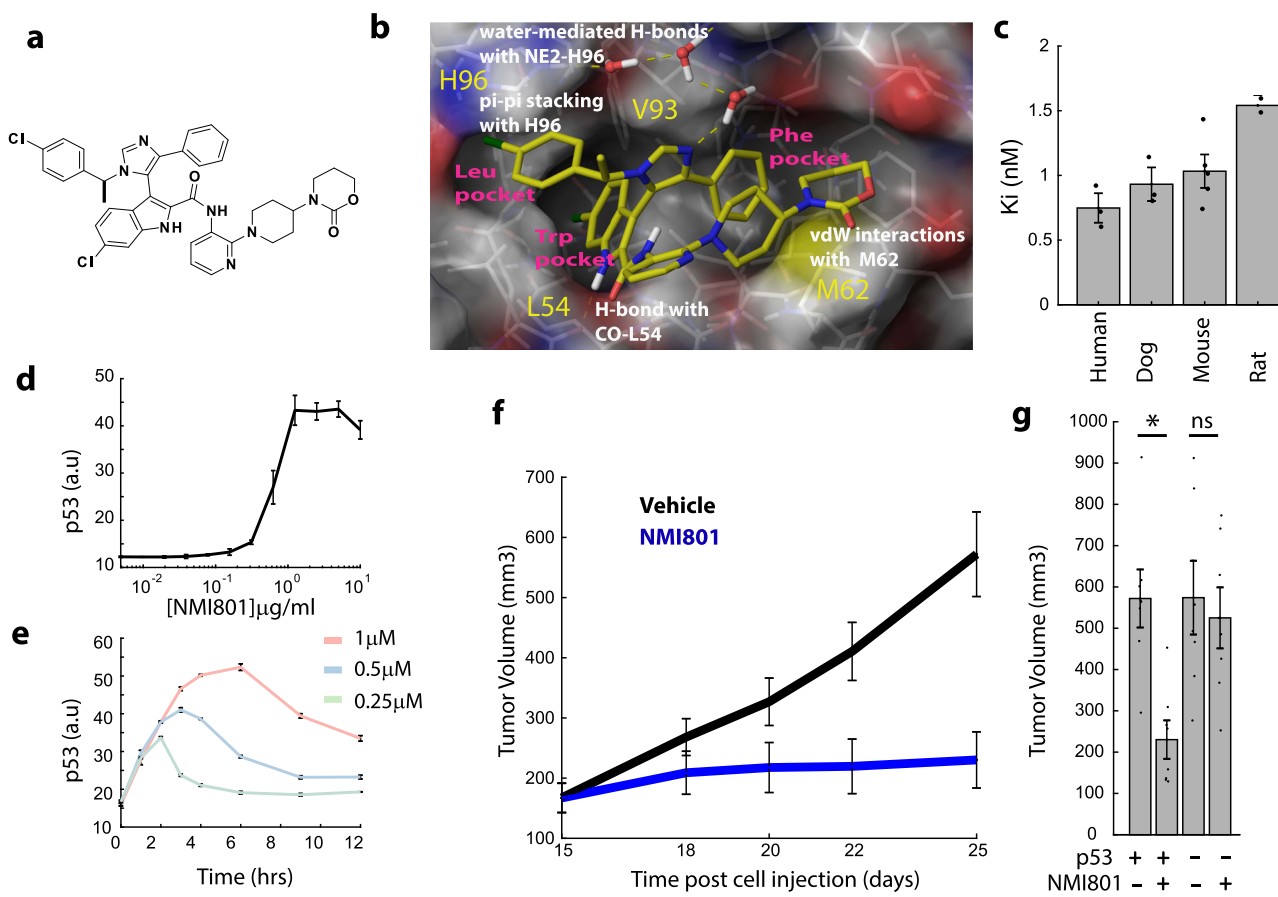

**Fig. 3 A novel Mdm2 inhibitor, NMI801, efficiently stabilizes p53 in mouse tissues and tumor xenograft models. a** Chemical structure of NMI801. **b** X-ray structure of NMI801 bound to the p53-binding pocket of Mdm2, at 2.1 Å resolution. H-bonds are indicated by dashed lines. The Mdm2 sub-pockets binding the p53 residues Leu 26, Trp 23, and Phe 19 are labeled Leu- Trp- and Phe-pocket, respectively. The coordinates have been deposited in the PDB databank (PDB access code = 6I29). **c** Ki of NMI801 for inhibiting p53-Mdm2 complex formation from a FRET competition assay using Cy5 labeled p53 peptide and full length Mdm2 from Human, Dog, Rat, and Mouse ($n = 3$ for human, dog; $n = 5$ for mouse; $n = 2$ for rat; error bars are SEM; points indicate individual measurements; mean is shown in bar plots). **d** Quantification of p53 levels by immunofluorescence in HEPA1C1C7 mouse cells 3 h after adding the indicated doses of NMI801 ($n = 4$ imaging fields; error bars are SEM). **e** Quantification of p53 levels by immunofluorescence in HEPA1C1C7 mouse cells at the indicated times after adding 0.25, 0.5, or 1 μM NMI801 ($n = 4$ imaging fields; error bars are SEM). **f, g** Xenograft tumors of HCT116 were engrafted for 15 days to an average size of ~150 mm³. Mice were then treated with vehicle or NMI801 daily (200 mg/kg). **f** Tumor volume was measured at the indicated times after NMI801 treatment. **g** Final tumor volume in p53 wild-type or null mice treated with vehicle or NMI801 ($n = 8$ mice/condition; error bars are SEM; dots indicate individual data points; *pval = 0.000356, two-sided $t$-test, no multiple comparison adjustment).

(Fig. 3f). Similar results were obtained with a second xenograft model of the human osteosarcoma cancer line SJSA-1 (Supplementary Fig. 5), suggesting the efficacy of NMI801 in both epithelial- and mesenchymal-derived tumors. Xenografts of isogenic p53 null derivatives (HCT116 p53$^{-/-}$) were unresponsive to NMI801, confirming the specificity of NMI801 for p53 (Fig. 3g). NMI801 was bioavailable and did not negatively affect the weight of the treated mice at doses sufficient to arrest tumor growth (Supplementary Fig. 6). Taken together these results confirm NMI801 as a potent Mdm2 inhibitor that stabilizes p53 in cells with compelling efficacy in vivo, highlighting its therapeutic potential.

We next tested the effect of the Mdm2 inhibitor NMI801 (Mdm2i) on p53 dynamics in vivo following radiation (IR). Specifically, we sought to determine if it can prevent the decrease in p53 levels and activity in the intestines in the hours after irradiation (Fig. 2a–e). We examined p53 levels at 5 h following radiation, a time at which p53 levels typically decline to baseline (Fig. 2a, c, d). We found that a combination of radiation and a single dose of Mdm2i maintained higher p53 abundance and

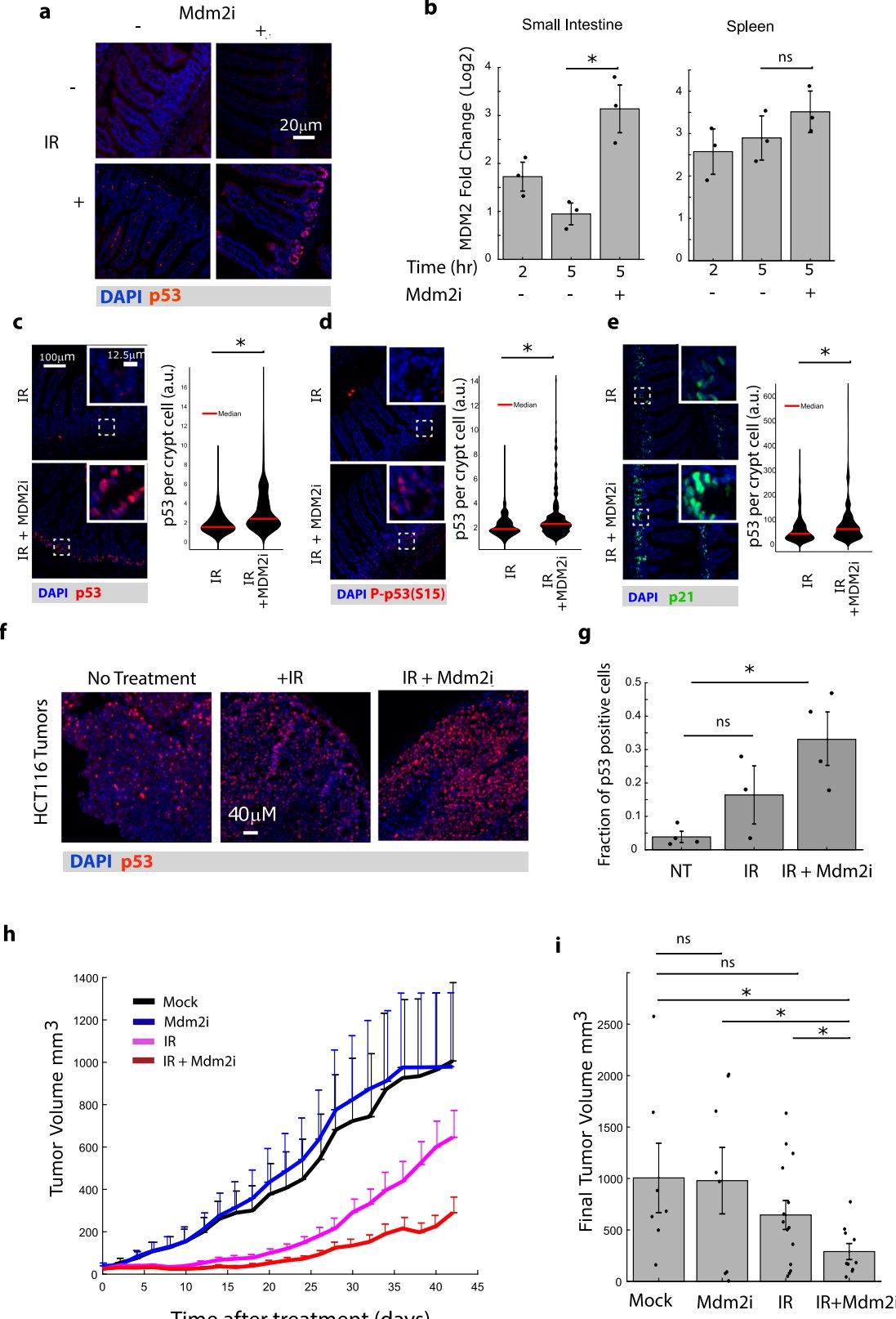

transcriptional activity (as measured by RNA levels of the p53 target gene MDM2) in the small intestine as compared to radiation alone (Fig. 4a, b). On the other hand, in the spleen, in which p53 abundance and activity were naturally sustained over 5 h, the Mdm2i had only a modest effect on p53 transcriptional activity (Fig. 4b). We further examined p53 abundance and

activity in individual crypt cells of the small intestine in irradiated mice treated with or without MDM2i. The abundance of p53 and its activated form, phospho-p53, were both increased in mice treated with MDM2i (Fig. 4c, d). Additionally, the expression of the p53 target gene p21 was also enhanced in crypt cells by treatment with Mdm2i (Fig. 4e), suggesting that the additional

**Fig. 4 Combination of radiation and Mdm2 inhibitor (Mdm2i) sustains p53 activity in tissues and tumors resulting in reduced tumor progression.** **a** Staining of p53 in the small intestine 5 h following the indicated treatments. Mdm2i was added 2 h after radiation (IR). Representative images from two independent experiments are shown. **b** p53 transcriptional activity in small intestine and spleen as measured by Mdm2 mRNA levels at the indicated time-points following treatment with Mdm2i. ($n = 3$ mice; error bars are SEM; *pval $= 0.0071$; two-sided $t$-test). **c–e** Representative images of staining for p53 (**c**), p53-phosphoS15 (**d**), and p53 target gene p21 (**e**) in the small intestines treated with IR alone or IR + Mdm2i as described in **a**. $n = 3$ mice, 15 crypts/mouse, *indicates significant, pval $= 1.7 \times 10^{-21}$ (**c**), $6.4 \times 10^{-9}$ (**d**), and $5 \times 10^{-5}$ (**e**) (two-sided $t$-test). **f** HCT116 tumors were engrafted for 21 days to an average size of 70 mm³ and then treated with radiation followed by vehicle or Mdm2i 2 h post radiation. Tumors were stained for p53 5 h after radiation treatment. Experiment was performed on 3–4 tumors/condition. **g** Quantification of data from **f** ($n = 3$–4 tumors per condition; dots indicate individual measurements; error bars are SEM; *pval $= 0.0053$; two-sided $t$-test). **h** Tumor sizes at the indicated time points in mice treated with or without radiation and then treated with or without Mdm2i 2 h post radiation (error bars are SEM). **i** Final tumor sizes from data in **h**. (* indicates significant, 0.0096, 0.0130, 0.0261; one-sided $t$-test). Dots represent individual tumors ($n = 7$–14 tumors/condition; error bars are SEM). No multiple test correction was used on the statistics in this figure.

p53 induced by Mdm2i was functional. Overall, these results suggest that Mdm2i effectively sustains functional p53 expression in the small intestine, which normally exhibits a transient p53 signal in response to irradiation.

We next investigated how sustaining p53 signaling using MDM2i after irradiation influenced the outcome of cultured cells and tumors to irradiation. The combinatorial treatment of irradiation (IR) + Mdm2i led to a reduction in cell viability of the colon cancer line HCT116, compared to either treatment alone (Supplementary Fig. 7). To test this therapy model in vivo, we established HCT116 xenograft tumors and compared p53 levels in response to ionizing radiation alone or in combination with Mdm2i. Recent data have shown that continuous dosing of Mdm2/4 inhibitors combined with radiation leads to improved tumor control[30]. Having established that continuous administration of Mdm2i inhibited growth of p53 wild-type tumors in the absence of radiation (Fig. 3f) and that following irradiation Mdm2i could sustain p53 levels and activity with a single dose (Fig. 4a, b), we next tested whether a single dose of Mdm2i shortly after radiation (2 h) could control tumor growth. We found that the combination of irradiation with Mdm2i increased p53 positive cells in the tumor when compared to radiation alone (Fig. 4f–g) and significantly suppressed tumor growth compared to either treatment alone (Fig. 4h, i). Taken together our results show that the Mdm2 inhibitor NMI801 switched p53's transient response post irradiation into a more sustained response and increases the radiosensitivity of cells even in tissues that are known to be more resistant to radiation (Fig. 5).

## Discussion

All tissues in our bodies and many tumors express a functional p53 protein, yet cells in different tissues respond differently to DNA damage and p53 activation[6,11,19,21,22]. Here we found a dynamic difference in p53 signaling between tissues. Following DNA damage, lymphoid organs (thymus and spleen) show sustained p53 levels and transcription of p53 target genes including the pro-apoptotic proteins PUMA. In contract, the small and large intestines show more transient activation of p53, specifically in the crypt region, and a corresponding transient expression of its target genes. Note that our results do not show significant differences in the identity of upregulated genes across tissues. Our focus on a small number of known p53 regulators of apoptosis (PUMA) and cell cycle arrest (p21) leave open the possibility that genome wide approaches would reveal more complex patterns of tissue-specific regulation by p53, as has been proposed by recent work on the diversity of p53 targets across cell types and tissues[31,32].

Antibody staining, as used in this study, allows association of specific markers at a given time-point, but is limited in connecting the history of events in a single cells. Further detailed studies connecting p53 dynamics with the dynamics of its target genes at the single-cell level in live cells in vivo will be necessary to understand how specific dynamical patterns of p53 lead to different cell fates in healthy and cancerous tissues. While our experiments do not show direct causality between p53 dynamics and radiosensitivity, our data shows a strong correlation between p53 dynamics and radiation sensitivity across tissues, as well as between chemically sustaining p53 dynamics and the activity of pro-cell cycle arrest and cell death signaling. Currently no genetic or chemical approach can decouple the time dynamics of p53 signaling from the average level of p53 protein. We believe that as our understanding of the p53 pathway improves, genetically engineered mouse models will be developed that can specifically perturb the time dynamics of p53 signaling to test its direct contribution to radiosensitivity in vivo.

The mechanisms leading to the differential dynamics of p53 between tissues are still unknown. The fact that Mdm2 is induced in the four tested tissues (Fig. 2) indicates that persistent p53 activation in tissues of lymphoid organs does not result from failure to activate the negative feedback between p53 and Mdm2. It is still possible that modification of Mdm2 activity, via post-translational modifications or differential feedback control of DNA damage signaling pathways, leads to the differential p53 dynamics. In addition, tissue-specific expression of other E3-ligases that are known to mediate p53 ubiquitination and degradation (e.g., Cop1, CHIP) may explain the observed tissue-specific temporal dynamics of p53 after DNA damage. Lastly, a tissue's ability to sense or repair the damage might be linked with the resulting p53 dynamics. For example, persistent unrepaired DNA damage may lead to sustained p53, while efficient and complete repair allows p53 to decline to its basal levels. Further studies and staining for additional regulators of the DNA damage and p53 pathways in different tissues are required to solve this open question.

The heterogeneity in p53 levels within tissues, especially in the small (jejunum) and large intestine (colon), is also poorly understood. Our data suggest that variations in p53 levels within each tissue do not result from different levels of radiation-induced DNA damage, but rather from differences in proliferative states, which are known to affect cell's sensitivity and respond to DNA damage as well as ability to repair. The striking restriction of p53 signaling to the basal crypt cells in the intestines, and the strong signaling in terminally differentiated but replication-competent hematopoietic cells, suggest that p53 responsiveness is not exclusively tied to stemness or proliferative state, but rather to additional, as yet unelucidated, feature of the cell. We expect single cell approaches, such as spatial RNA sequencing, will begin to reveal the source of this heterogeneity.

Modification of p53 behavior with small molecules has been a long-term goal of therapeutic approaches to both directly kill tumor cells and minimize toxicity in healthy cells by suppressing p53 signaling[16,18]. For example, it has been suggested that

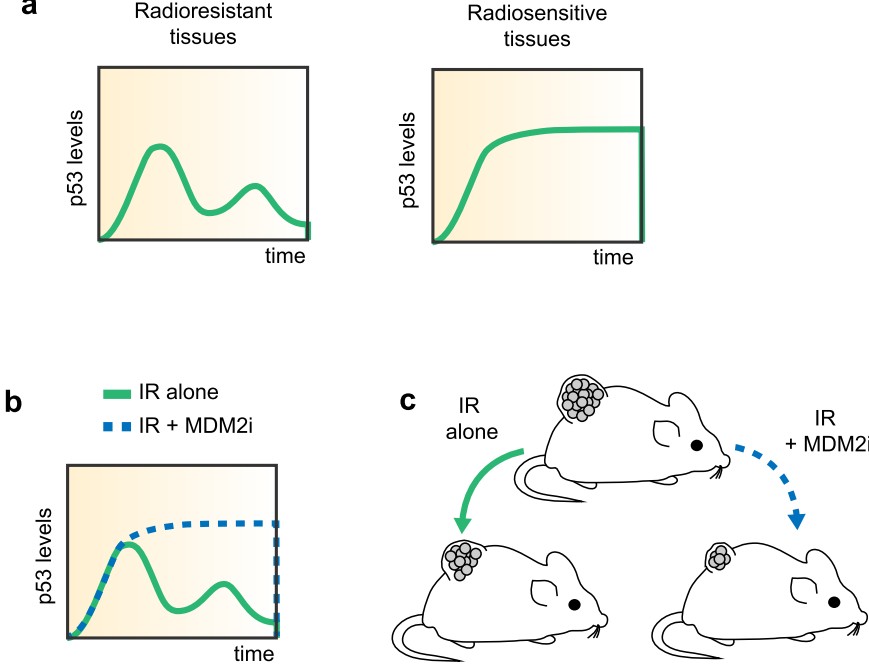

**Fig. 5 Schematic of the proposed mechanism for the control of radiation sensitivity of tissues and tumors via p53 dynamics. a** Radioresistant tissues, such as the small and large intestines, show transient induction of p53 and oscillatory behavior of its activity and target genes, while the more sensitive tissues, such as the spleen and thymus show sustained levels of p53 after irradiation (IR). **b, c** Combination of radiation (IR) with the Mdm2 inhibitor NMI1801 (Mdm2i) alters p53 dynamics (**b**) and suppresses tumor growth (**c**).

inhibitors of p53, or its downstream targets such as PUMA, might provide a mechanism to spare sensitive normal tissues from the effects of chemotherapy[33,34]. Recently a number of clinical trials have tested the safety of single agent MDM2 inhibitors[16,17], and more recently trials have launched combining DNA damaging agents and MDM2 inhibitors[19,35,36]. These trials have relied on the synergy between sustained MDM2 inhibition and radiation or chemotherapy. However, these ongoing trials are not designed to examine the mechanisms leading to the synergistic effect nor the timing between the two treatments. Here we show that a single dose of MDM2 inhibitor delivered shortly after radiation treatment of a tumor results in improved tumor control. Our results therefore argue for the importance of developing strategies to manipulate the dynamics of p53 in vivo for tuning the specificity of cancer therapy. In addition, our work offers an important first step in investigating p53 dynamics in vivo and determining the connection between its dynamics and tissue sensitivity.

## Methods

**Cell lines**. Parental HCT116 cell lines were obtained from ATCC. Their identity was verified with STR profiling. HCT116 p53 null cells and their corresponding control line were a gift from the Vogelstein lab, John Hopkins Medicine. HEPA1C1C7 were a gift from the Weitz lab, Harvard Medical School. SJSA-1 cell lines were obtained from ATCC.

**Immunofluorescence of cell cultures**. Cells were plated in 96-well flat-bottom plates (Corning), grown for 24–48 h, treated as indicated with NMI801 dissolved in DMSO to a stock concentration of 10 uM. Cells were fixed at the indicated times with 2% paraformeldahyde (PFA, Alfea Asear) for 10 min. Plates were washed twice with Phosphate Buffered Saline (PBS) and permeabilized with 0.1% Triton X-100, before sequential staining with primary and secondary antibodies. Cells were washed three times with PBS and imaged within 24 h.

**Quantification of cell survival in culture after IR or MDM2i treatment**. HCT116 cells ($2.5 \times 10^5$) were plated in 10 cm plates, incubated for 24 h and then treated with irradiation alone (5gy – Cobalt Gamma source), Mdm2i (NMI801, 1 μM, 24 h) alone, or a combination of radiation and Mdm2i (MDM2 inhibitor was

added 2 h after IR treatment, and removed 24 h after treatment to mimic drug treatment and excretion in the animal experiments). Two days after treatment cells were split, counted, and plated at multiple dilutions in 12 well plates. Plates were fixed and imaged after 7 days and colonies counted manually.

**Antibodies and reagents for cell culture**. Primary antibodies used were: p53 (CM5, Leica) 1:1000 dilution, p53-s15p (#9284, Cell signaling) 1:1000 dilution, FL393 (Santa Cruz) 1:400 dilution, pH2AX (ab11174, abcam, 1:400 dilution). Secondary goat anti-mouse or anti-rabbit antibodies conjugated to AF555 or AF647 were purchased from Invitrogen and used at 1:1000 dilution.

**Histology and antibodies**. Organs from C57BL/6 and HCT116 tumors from Nu/Nu mice were harvested, fixed in 10% formalin solution, and paraffin-embedded. After deparaffinization and rehydration of the tissue, antigen retrieval was performed using BD Retrievagen A (pH 6.0) (BD Biosciences). The sections were blocked with 4% normal goat serum in PBS at room temperature for 1 h and primary antibodies.

Primary antibodies were p53: CM5 (Leica, 1:1000 dilution) for mouse tissue, p53: 7F5 (Cell Signaling) for tumors. Gamma H2A.X (phospho S139, 1:1000 dilution) was stained with Abcam antibody EP854(2)Y and H2AX phospho – AF488 (2F3, Biolegend, 1:25 dilution) for co-staining with p53. Olfm4 (D6Y5A) was directly conjugated to AF488 (Cell Signaling, 1:25 dilution) for co-staining with p53. Proliferation was measured by MKi67-FITC (SOLa15, Thermofisher, dilution 1:25). Apoptosis was detected with anti-Cleaved Caspase-3 (cell signaling #9661, 1:100 dilution). The p53 target gene p21 (Abcam, EPR362, 1:100 dilution). Phospho-p53 (S15) was stained with a polyclonal antibody from Invitrogen (PA5-104741, 1:50 dilution).

All primary antibodies were incubated overnight at 4 °C. A goat anti-rabbit IgG (H + L) secondary antibody (ThermoFisher Scientific) was applied at 1:100 dilution and incubated at room temperature for 1 h. The sections were counterstained with DAPI (4′,6-Diamidio-2-Phenylindole, Dyhidrochloride) (ThermoFisher Scientific) to identify nuclei and the images were captured and analyzed as described above. According to manufacturer's instruction, DeadEnd Fluorometric TUNEL System (Promega) was used to detect apoptotic cells.

**Microscopy and image analysis**. Fixed microscopy was performed using a Nikon TI or TI2 microscope equipped with an epi-fluorescent source (either mercury arc lamp (Prior) or LED system (Lumencor)), automated stage, FITC, Cy3, Cy5, and DAPI filter sets (Chroma) and CCD or CMOS camera (Hamamatsu or Photometrics). Slides were imaged as a single large-stitched object, each frame was background and flat field corrected. Images were stitched into tissue level views which were then processed with custom Matlab code to obtain single cell intensities

for p53 or g-H2AX staining. Briefly, our code segments nuclei using the DAPI channel and applies this mask to the p53, g-H2AX, or TUNEL channels. Cells with excess intensity in an unstained channel (FITC or TRITC) were removed as autoflorescent (often red blood cells). Imaging for time-course data or paired samples (e.g., treated/untreated) was done over the course of 1–2 days using identical imaging settings. For evaluation of cell location relative to the bottom of the crypt in the small intestine, the location of each cell was manually annotated and its p53 intensity was scored. For xenograft samples, the tumor section were segmented using the DAPI channel, p53 per cell signal was normalized to DAPI to correct for illumination effects and fraction of p53 cells were called using a uniform cutoff across samples ($N > 1000$ cells/tumor). Further details on the specific analyses are given below.

**Image analysis: small intestine for p53, Ki67, and γH2AX**. Tissues were stained with combination for p53 (CM5) and then with a secondary anti-rabbit 647 antibody (Invitrogen), followed by staining for γH2AX or Ki67 using primary conjugate antibodies with Alexa488. The tissue sections were then imaged and processed to detect nuclei using an automated matlab script (Mathworks). Crypt structures were then identified manually and the bottom most cell identified. For each crypt 20–70 cells were then counted from the bottom of the crypt to the top of the villi (in the case of γH2AX) or to the midpoint of the Villi (for Ki67 which is crypt resident). For each sample 6–18 crypts were quantified in this way. The fluorescent value of each identified cell was quantified. For p53 signals, where the signal intensity is weak relative to the background, the raw intensity value was divided by the control channel to compensate for uneven illumination and autofluorescence. The signal of each cell for p53 and Ki67/γH2ax was then plotted against its position in the crypt (Fig. 1).

**Image analysis: p53 and P-p53 multi-tissue comparison**. All tissues were processed for histology in parallel. Tissues stained with the appropriate antibodies were imaged with the same microscope settings and processed to detect nuclei using an automated matlab script (Mathworks). Signal of each tissue was quantified by measurement of the 10 brightest pixels within the nucleus for each channel (DAPI, Secondary Antibody, and Control). For P-p53 (S15) and p53 signals the raw intensity value was divided by the control channel to compensate for uneven illumination and auto-fluorescence. As p53 is active in only a small subset of the cells from each tissue we estimated the p53/P-p53 signal as the average of the top 2% of cells for the small intestine or large intestine (our estimate of the crypt population). For the spleen and thymus we took the average in the top 10% of cells. The results we observe were robust to modest changes in these values.

**Image analysis: small intestine for p53, P-p53, and p21 signals with and without drug treatment**. Tissues stained with the appropriate antibodies were imaged with the same microscope settings and processed to detect nuclei using an automated matlab script (Mathworks). Signal of each tissue was quantified by measurement of the 10 brightest pixels within the nucleus for each channel (DAPI, Secondary Antibody, and Control). Crypt structures were then identified manually and six cells counting from the bottom of the crypt were selected in 15–20 crypts per tissue slice. The fluorescent value for each of these cells was quantified (top 10 pixel in the nucleus). For P-p53 and p53 signals, where the signal intensity is weak relative to the background, the raw intensity value was divided by the control channel to compensate for uneven illumination and auto-fluorescence. Cells were pooled from three mice for each condition and compared.

**qPCR**. Tissue samples were preserved in RNAlater (Ambion) immediately after animal sacrifice and stored at −20 or −80 °C. Subsequently ~5 mg of tissue was diced with a razor blade and placed into trizol (Ambion) or Quizol (Quiagen), and the sample was vigorously mixed by pipetting and vortexing to dissociate the tissue. RNA was purified by phenol chloroform following the trizol RNA purification protocol (Ambion) and eluted in nuclease free water (Ambion). 1–2 ug of RNA samples were reverse transcribed using the high capacity reverse transcription kit (Applied Bio Sysems) according the manufacture's protocol. 8 ng of cDNA was used for each qPCR reaction. Primers to ACTB, CDKN1A, MDM2, or BBC3 were used for a RT-QPCR run on a CFX96 (Biorad) instrument using Sybr-green master mix (Invitrogen). All reactions were performed in duplicates and averaged.

**Primers:**

| | | |
|---|---|---|
| ACTB | ACCTTCTACAATGAGCTGCG | CTGGATGGCTACGTACATGG |
| MDM2 | GCGTGGAATTTGAAGTTGAGTC | CTGTATCGCTTTCTCCTGTCTG |
| CDKN1A | CAGATCCACAGCGATATCCAG | AGAGACAACGGCACACTTTG |
| BBC3 | CTGGAGGGTCATGTACAATCTC | GGTGTCAGAAGGCGGAG |

**X-ray crystallography for MDM2i/NMI801**. The protein solution for MDM2 Gly-(S17-N111) (numbering according to Q00987) was 8 mg/mL in 50 mM TRIS pH 8.0, 200 mM NaCl, 1 mM TCEP, 10% glycerol, and the complex with NIM801

was obtained by adding a 3-fold molar excess of compound. Co-crystals were obtained at 20 °C and by sitting drop vapor diffusion. The drops were made up of 400 nL of protein solution and 400 nL of well solution and the reservoir solution consisted of 2.1 M AmSO4, 0.1 M CitricAcid, pH 3.5. All crystals were cryo-protected in well solution supplemented with 15% glycerol and flash frozen in liquid nitrogen. X-ray data were collected at the Swiss Light Source SLS, Villigen, Switzerland) and data processing was done with XDS[37]. The structure was determined by molecular replacement (PHASER(25)) using 4DIJ as a search model. Programs REFMAC[38] and COOT[39] were used for refinement and model (re)building. The final refined structure has an R (Rfree) value of 0.243 (0.277) and showed all residues in the preferred regions of the Ramachandran plot. The X-ray figure was prepared with PyMOL[40].

**MDM2 affinity measurements by Time Resolved Fluorescence Energy Transfer (TR-FRET) assay**. To measure the ability of Mdm2i (NMI801) to inhibit p53 binding to MDM2 we utilized a competitive binding assay with a FRET readout. The measurements were performed in white 1536-well microtiterplates (Greiner Bio-One GmbH) in a total volume of 3.1 ul combining 100 nl of compounds diluted in 90% DMSO/10% H2O (3.2% final DMSO concentration) with 2 μl Europium-labeled streptavidin (final concentration 2.5 nM) in reaction buffer (PBS, 125 mM NaC1, 0.001% Novexin (Novexin Ltd.), Gelatin 0.01%, 0.2% Pluronic (BASF), 1 mM DTT), followed by the addition of MDM2-Bio diluted in assay buffer (final concentration 10 nM). The solution was pre-incubated at room temperature for 15 min, followed by addition of 0.5 ul Cy5-p53 peptide in assay buffer (final concentration 20 nM). The sample was incubated at room temperature for 10 min prior to reading the plate. Europium-Cy5 FRET was measured by an Analyst GT multimode microplate reader (Molecular Devices) with the following settings: dichroic mirror 380 nm, excitation 330 nm, emission donor 615 nm, and emission acceptor 665 nm. To validate the TR-FRET assay, a reference compound was added in each run. Only if the reference compound scored in the expected potency range, the data was considered reliable. IC50 values are calculated by curve fitting using XLfit. If not specified, reagents are purchased from Sigma Chemical Co, St. Louis, MO, USA.

**Xenograft studies using NMI801**. All animal studies were conducted in accordance with ethics and procedures covered by permit nos. BS-1975 issued by the Kantonales Veterinäramt Basel-Stadt and in strict adherence to guidelines of the Eidgenössisches Tierschutzgesetz and the Eidgenössische Tierschutzverordnung, Switzerland. All animals had access to food and water ad libitum and were identified with transponders. They were housed in a specific pathogen-free facility with a 12-h light/12-h dark cycle. HCT116 or SAJA cells obtained from ATCC were injected subcutaneously at a concentration of $3 \times 10^6$ cells/100 microliters in the right flank of Harlan nude mice. Two weeks later, mice were treated daily with NMI801 at 200 mg/kg or vehicle (20% Microemulsion Pre-Concentration No. 5, 80% water). Mouse body weight and tumor volume were measured every 2 to 3 days. Pharmacokinetics in flash-frozen tissues were measured 3 h post first treatment in a separate cohort of mice and 3 h post last treatment on the last day. The concentration of NMI801 was measured by UPLC/MS–MS assay.

**Radiation treatment**. Dual source 137Cs Gammacell 40 Exactor (Best Thera-tronics) was used to irradiate all mice. Whole body irradiation was delivered to unsedated mice. For localized radiation, the same source was used with a custom-built conformal tumor irradiation lead shield, which has been described previously[41]. Prior to irradiation, 87.5 mg/kg ketamine and 12.5 mg/kg xylazine were administered by i.p. injection. Mice were immobilized in the lead-shielding chamber, and animals were irradiated at the calibrated 0.6Gy min-1 dose rate for a total of 10 Gy.

**Xenograft studies using NMI801 and IR combinations**. Animal research was performed in accordance with guidelines from the Institutional Subcommittee on Research Animal Care and with approval of the Institutional Animal Care and Use Committee at the Massachusetts General Hospital Research Institute. HCT116 cells were injected subcutaneously at a concentration of 1.0E + 06/100 ul in the right and left flank of Nu/Nu mice (30 mice in two batches, Charles River). Tumors (2 per mouse) were allowed to vascularize and grow for 13–16 days, reaching an average size of ~16 mm³. Tumors were treated with ionizing radiation selective for the tumor site as described above. Mice were treated with 200 mg/kg NMI801 dissolved in a solution of 2% methyl cellulose and 0.8% Tween-80 by gavage. Tumor dimensions were measured by caliper every 2–3 days for 42–45 days after treatment. Tumors that did not have a measurable volume were discarded. If a mouse was sacrificed its final tumor size was fixed until the end of the experiment. To merge the results of two separate mouse experiments, tumor size was linearly interpolated to match the measurement days. For acute radiation treatment (Fig. 4h, i) tumors were allowed to develop for 3 weeks to an average size of ~70 mm³ before treatment with IR and NMI801 as described above. Mice were than sacrificed and tumors excised at 5 h after irradiation.

**Normal tissue measurements**. Animal research was performed in accordance with guidelines from the Institutional Subcommittee on Research Animal Care and

with approval of the Institutional Animal Care and Use Committee at the Massachusetts General Hospital Research Institute. Mice (C57BL/6) were irradiated with a gamma source in batches of 2–6, numbers indicated in each experiment. They were subsequently sacrificed and organs harvested and preserved in PFA solution for histology or RNAlater (ambion). Tissues were paraffin embedded and 5 µm slices cut. Tissues were stained using primary and secondary antibodies using standard procedures as described under histology.

**Statistics**. No predetermined sample sizes were used. Authors were not blinded prior to analysis. Samples were compared with one- or two-tailed $t$-tests with a 0.05 cutoff for significance as noted in the figure legends.

**Reporting summary**. Further information on research design is available in the Nature Research Reporting Summary linked to this article.

## Data availability

The main data supporting the findings of this study are available within the article and its supplementary materials. Source data for all figures are provided with this paper. Additional data and materials are available from the corresponding authors upon request, except for NMI801, which is an internal research compound patented by Novartis Institute for Biomedical Research. The structural data of NMI801 is available from the protein data bank at doi:10.2210/pdb6I29/pdb and the details how to synthesize it are in patent WO2012/176123[29]. Source data are provided with this paper.

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

## Acknowledgements

We thank members of the Lahav and Weissleder laboratories for helpful comments and suggestions throughout this work. We thank Jose Reyes for the illustration in Fig. 5.

## Author contributions

J.S.O., Y.I., M.A.M., S.F., P.H., J.K., P.F., W.C.F., R.W. and G.L. designed the experiments. J.S.O., Y.I., M.A.M., M.A.P., S.F., P.H., J.K. and P.F. preformed the experiments. J.S.O., Y.I., M.A.M., S.F., P.H., J.K., P.F., W.C.F., R.W. and G.L. analyzed the data. J.S.O., A.J., W.C.F., R.W. and G.L. wrote and edited the manuscript.

## Funding

This work was supported in parts by NIH grants GM116864, GM083303, CA207727, CA206997, CA206890, funding from the Novartis Institute for Biomedical Research and the Ludwig Center at Harvard.

## Competing interests

The authors declare no competing interests.
