## [Peer Review File · Nature Communications]

Reviewers' comments:

Reviewer #1 (Remarks to the Author): Expertise in in vivo imaging

This manuscript tries to link differences in the dynamics of p53 in vivo to the response to radiation. This is very important for human health, and to start evaluating how differences in transcription factor dynamics often described in cell culture assays affect whole tissues and organs in vivo.

The main issue is that the paper falls short of establishing a clear relationship between in vivo dynamics of p53 and the response of tissues and organs to radiation. I recognize that this is a very challenging open problem. But at the moment, the main take home message from the paper is that manipulating p53 levels in vivo affects radiation sensitivity. I fail to see how the paper can directly establish that the dynamics of the protein are functionally important. It is not surprising that p53 levels display a peak after radiation, and that artificially maintaining high p53 levels using a drug can affect radiation sensitivity.

Additional points:

- Figure 1. The results need quantification, currently they are only qualitative descriptions.

Can they co-stain for p53 and gH2AX, quantify the results and establish that indeed, differences in p53 levels are uncorrelated to DNA damage levels? By eye, there seems to be extensive cell-cell variability throughout the tissues, not only in p53 but also gH2AX. Shouldn't figures 1 and 2 be combined?

- All microscopy images could be a lot bigger. There is lots of white space between figure panels.

- Supp Fig 1 is very important, why are these data in supplemental materials? The images are displayed in an unsatisfactory manner. The levels of p53 in the large intestine are very difficult to examine. And in the spleen, the images show very different background level.

These experiments need to be more thoroughly analyzed. The levels of p53 should be compared using the same imaging conditions in all images, and the fluorescent levels normalized to DAPI or other fluorescent signals, which are not expected to change as a consequence of IR. For example, in

Fig. 2A, p53 levels are higher in the bottom part of the image, but so is DAPI. The methods section does not describe well enough how p53 and other proteins levels are quantified and compared.

- Why is the word 'dynamics' written in italics in the text?

- Check for grammar, typos in: Mdm2 inhibitors are class of molecules currently being evaluated in clinical trials for tumor therapy¹⁶⁻¹⁸.

- The last part of Introduction describes the results, this is unnecessary.

Reviewer #2 (Remarks to the Author): Expertise in targeting p53

The manuscript by Ornstein et al addresses the dynamics of p53 in different tissues of gamma-irradiated (IR) mice and its coupling with radiation sensitivity. This lab has been pursuing studies on p53 dynamics for quite a while, with first publication in 2012. Since then, they have published several studies addressing different aspects of p53 dynamics in cells. In the current study, they used p53 staining at different time points after IR of mice to describe the differential oscillation of p53 depending on tissue. The authors linked this differential oscillation of p53 with differential tissue sensitivity to IR. Their experiments suggest that it is possible to radiosensitize tumor cells in vivo by single-dose treatment with MDM2 inhibitor to prevent p53 oscillation. This is a clear and a very straightforward study, with some very interesting observations.

However, I would expect this group, who had studied p53 dynamics in so much detail, to provide a deeper, mechanistic insight. The differential p53 induction coupled with differential response of tissues as well as differential induction of p53 target genes in irradiated mice has been reported before in a number of publications, cited in this manuscript. The novelty of this study is that the differential induction of p53 is explained by differential oscillation. But is it really an explanation? It remains completely unclear, why p53 oscillations depend on a tissue type; which genes are induced upon short versus long oscillations, why cells do not die upon short oscillations? Why p53 is not induced in intestine cells with high level of gamma H2AX, the evidence of unrepaired DNA damage? The authors have not addressed any of these questions.

The idea of combination of IR with MDM2 inhibitors is also not new, it has been described in publications and even being tested in clinical trial. The finding that a single dose of MDM2i can have a dramatic tumour suppressive effect in combination with IR is exciting. However, in other publications (ref # 27, for example), 3 doses of MDM2i were not very efficient, but 10 doses were. Thus, it depends very much on the type of MDM2i. To show that the novel MDM2i, which the authors introduce in this study, is indeed having this fantastic effect, it should be tested in another tumour models. What happens to normal tissues, such as intestine, upon MDM2i and IR combination?

Other comments:

Error bars and significance of observed differences should be shown in Fig 2e, Suppl Fig2, Suppl Fig3, Fig 4e,f

Figure legends are not adequate in many cases. For example, it is not indicated in Fig 4, which doses of MDM2 and IR were used? The method for measurement of binding of MDM2i to p53 peptide is not described in Figure legend, nor in Results section. No controls have been provided for this experiment either.

References #7 and 21 are the same, as well as ref #8 and 23

Reviewer #3 (Remarks to the Author):Expertise in p53 (in vivo) and radiosensitivity

In “p53 dynamics vary between tissues and are linked with radiation sensitivity” Ornstein et al examined the effects of total body irradiation on various tissue types and the p53 response in those tissues. The authors examined the p53 pathway over a 7-hour time course in the thymus, spleen, and large and small intestines. In the small intestine, they broke down the expression of various pathway members by cell position as well. They further showed that a novel MDM2 inhibitor was able to induce p53 and reduce xenograft tumor growth relative to vehicles. However, a more expansive analysis of these pathways by cell position both before and after MDM2i treatment is needed.

Major Criticism:

Methods:

1. Please state whether the cell position includes Lgr5+ CBCs in the small intestines. If not, it is necessary to include Lgr5+ CBCs.
2. Please include unirradiated controls for TUNEL stained slides in Figure 1.

Figures:

1. Figure 1C: Please show staining for cleaved caspase-3 in addition to TUNEL to elucidate p53-dependent apoptosis vs. cell death in general.
2. Figure 2: Please also stain tissues for phospho-p53, p21, MDM2, PUMA, cleaved-caspase 3, TUNEL and BrdU. Please quantify these in the manner of figures 2B, 2C, and 2E.
3. Figure 2: Please clarify whether, for small intestine cell positions, n>100 cells is total or per position.
4. Figure 2: Please repeat the experiments in this Figure using p53^{-/-} mice.
5. Figure 4: Please repeat the staining of the small intestines suggested in Figure 2 in mice treated with IR +/- Mdm2i to elucidate mechanisms and effects of MDM2i on normal tissue response to DNA damage.

Discussion:

1. Please address what makes this novel MDM2 inhibitor different from others in use. Does it have unique biochemical properties that allow for it to be administered as a single dose, or is that simply an untested method?

Minor Criticism:

1. In all of your figures of stained tissues, please avoid red and green coloring.

Response to Reviewers' comments:

Reviewer #1: *This manuscript tries to link differences in the dynamics of p53 in vivo to the response to radiation. This is very important for human health, and to start evaluating how differences in transcription factor dynamics often described in cell culture assays affect whole tissues and organs in vivo.*

We thank the reviewer for his/her general interest in our work and for emphasizing the importance of the questions addressed in this work.

The main issue is that the paper falls short of establishing a clear relationship between in vivo dynamics of p53 and the response of tissues and organs to radiation. I recognize that this is a very challenging open problem. But at the moment, the main take home message from the paper is that manipulating p53 levels in vivo affects radiation sensitivity. I fail to see how the paper can directly establish that the dynamics of the protein are functionally important. It is not surprising that p53 levels display a peak after radiation, and that artificially maintaining high p53 levels using a drug can affect radiation sensitivity.

We apologize that the novelty of our work was not clear. The dynamics of p53 and its impact on cell fate have indeed been previously investigated in cell cultures. However, p53 dynamics in vivo and its potential impact in normal and cancerous tissues have not been demonstrated. One take home message, as the reviewer noted, is that manipulating p53 in vivo affects radiation sensitivity. Importantly, we showed that a combination of IR with a *single* dose of MDM2i alters p53 dynamics and suppresses tumor growth. In addition, our measurements in normal tissues show for the first time that p53 dynamics vary between healthy tissues, and that this variation has functional consequences in terms of the dynamics of downstream gene expression. We agree that a direct link between p53 dynamics and cell fate *in-vivo* can be achieved by measuring these in live cells. We are in the process of establishing live-cell reporters for intravital imaging in mice. These experiments are challenging and it will take a few more years until we are able to fully execute them. We revised the text to better highlight the novelty of the work, discuss its limitation and the future work required to overcome them.

Additional points:

- *Figure 1. The results need quantification, currently they are only qualitative descriptions. Can they co-stain for p53 and γ H2AX, quantify the results and establish that indeed, differences in p53 levels are uncorrelated to DNA damage levels? By eye, there seems to be extensive cell-cell variability throughout the tissues, not only in p53 but also γ H2AX.*

We Agree. Figure 1 includes examples of images with the goal of presenting our *in-vivo* raw imaging data. We have now added quantification of our time-series imaging data for all tissues (**new Figure 2a**). In addition, to address the reviewer's concern about γ H2AX, we have co-

stained for p53 and γ H2X and quantified them in the small intestine (**new Figure 1c, d**). This direct comparison revealed that p53 and γ H2AX are modestly anti-correlated; γ H2AX levels were high in the villi and low in the crypts, while p53 showed the opposite trend. We suspected that proliferative state may contribute to this anti-correlation between γ H2AX and p53 levels and therefore co-stained for p53 and Ki67, a marker for cycling cells. Indeed we found a substantial overlap between p53-positive and Ki67-positive cells in the crypts (**new Fig. 1e-g**). Replicative state is known to impact the kinetics of DNA repair and the choice of repair mechanism, which can explain the relative lower levels of γ H2AX in cycling crypt cells. We added these new figures and explanation to the main text.

- Shouldn't figures 1 and 2 be combined?

Similar to the reviewer, we have also considered that. However, due to the large number of images and in light of new data added in this revision, we chose to keep them separated.

- All microscopy images could be a lot bigger. There is lots of white space between figure panels.

We thank the reviewer for this suggestion and have increased the size of the images while preserving an appropriate figures size. We also minimized the white spaces between images.

- Supp Fig 1 is very important, why are these data in supplemental materials? The images are displayed in an unsatisfactory manner. The levels of p53 in the large intestine are very difficult to examine. And in the spleen, the images show very different background level. These experiments need to be more thoroughly analyzed. The levels of p53 should be compared using the same imaging conditions in all images, and the fluorescent levels normalized to DAPI or other fluorescent signals, which are not expected to change as a consequence of IR. For example, in Fig. 2A, p53 levels are higher in the bottom part of the image, but so is DAPI. The methods section does not describe well enough how p53 and other proteins levels are quantified and compared.

The reviewer is correct and we apologize for not providing sufficient information about our image analysis and quantification. There is indeed variation in the fluorescence signals, as well as the background across a tissue slice. While we cannot eliminate this variation experimentally, it is important to point out that all images in a given panel were acquired under similar conditions and microscope settings. To address this variation, our quantitative image analysis included background subtraction and normalization to DAPI or FITC. We have now added several new paragraphs to the methods section explaining our image acquisition and analysis approaches. In addition, our main findings from quantifying the tissue images are strongly supported by qPCR (**Figure 2g**). Note that we now show more examples of images and their quantifications in the main figures (**Figure 1, 2**). Due to the large number of data, we kept the time-series images in the Supplementary Materials (**Figure S4**).

- Why is the word 'dynamics' written in italics in the text?

The word 'dynamics' was written in italics only once in the text for emphasis.

- Check for grammar, typos in: *Mdm2 inhibitors are class of molecules currently being valuated in clinical trials for tumor therapy*16-18.

We have corrected this sentence and have carefully rechecked the manuscript for grammar and typos.

- *The last part of Introduction describes the results, this is unnecessary.*

We think that a brief description of the main findings at the end of the introduction is helpful for readers, and therefore chose to keep it. We are open to removing it if recommended by the editor.

Reviewer #2: *The manuscript by Ornstein et al addresses the dynamics of p53 in different tissues of gamma-irradiated (IR) mice and its coupling with radiation sensitivity. This lab has been pursuing studies on p53 dynamics for quite a while, with first publication in 2012. Since then, they have published several studies addressing different aspects of p53 dynamics in cells. In the current study, they used p53 staining at different time points after IR of mice to describe the differential oscillation of p53 depending on tissue. The authors linked this differential oscillation of p53 with differential tissue sensitivity to IR. Their experiments suggest that it is possible to radiosensitize tumor cells in vivo by single-dose treatment with MDM2 inhibitor to prevent p53 oscillation. This is a clear and a very straightforward study, with some very interesting observations.*

We thank the reviewer for their interest in our questions and observations.

However, I would expect this group, who had studied p53 dynamics in so much detail, to provide a deeper, mechanistic insight. The differential p53 induction coupled with differential response of tissues as well as differential induction of p53 target genes in irradiated mice has been reported before in a number of publications, cited in this manuscript. The novelty of this study is that the differential induction of p53 is explained by differential oscillation. But is it really an explanation? It remains completely unclear, why p53 oscillations depend on a tissue type; which genes are induced upon short versus long oscillations, why cells do not die upon short oscillations?

In this work we showed, for the first time, that p53 dynamics vary between tissues *in-vivo*, and that manipulation of its dynamics changes cell fate. The open questions that the reviewer is raising are all important and interesting. However, addressing the mechanisms leading to p53 differential dynamics across tissues and the mechanism translating these dynamics into specific cellular outcomes will require significant additional investigation *in-vivo*, which is beyond the

scope of this work. To address the reviewer's comment, we now discuss the open questions that emerge from our work and the additional work required to address them.

Why p53 is not induced in intestine cells with high level of gamma H2AX, the evidence of unrepaired DNA damage? The authors have not addressed any of these questions.

This is an important question, similar to the one raised by Reviewer 1. To address this we co-stained p53 with γ H2X and with Ki67. This direct comparison showed that indeed in intestine cells p53 and γ H2AX are modestly anti-correlated (**new Figure 1c, d**). We hypothesized that active repair mechanisms in the cycling crypt cells allow them to repair DNA damage more rapidly and completely than in enterocytes. Indeed we found a substantial overlap between p53-positive and Ki67-positive cells in the crypts (**new Fig. 1e-g**). Replicative state is known to impact the kinetics of DNA repair and the choice of repair mechanism, which can explain the relative lower levels of γ H2AX in cycling crypt cells. We added these new figures and explanation to the main text.

The idea of combination of IR with MDM2 inhibitors is also not new, it has been described in publications and even being tested in clinical trial. The finding that a single dose of MDM2i can have a dramatic tumour suppressive effect in combination with IR is exciting. However, in other publications (ref # 27, for example), 3 doses of MDM2i were not very efficient, but 10 doses were. Thus, it depends very much on the type of MDM2i. To show that the novel MDM2i, which the authors introduce in this study, is indeed having this fantastic effect, it should be tested in another tumour models.

This is an important point. We now include data from a second model using xenograft of SJSA-1 tumor cells (**New Supplementary Figure 5**). Our data are consistent with the data acquired with HCT116 xenograft, demonstrating that the MDM2 inhibitor NMI801 is effective in tumors of both epithelial and mesenchymal origin

What happens to normal tissues, such as intestine, upon MDM2i and IR combination?

This is an interesting question. We now show that in the small intestine combination of the MDM2 inhibitor with IR results in sustained p53 and phospho-p53 (**new Figure 4 c, d**) as well as in an increase in the expression of p53 canonical target gene p21 (**New Figure 4e**).

Other comments:

Error bars and significance of observed differences should be shown in Fig 2e, Suppl Fig2, Suppl Fig3, Fig 4e,f

We have added error bars and p-values to the figures and included new quantifications of time-course histology with triplicate measurements at 0,2,5,7 hrs to strengthen our analyses and findings (**New Fig. 2a**).

Figure legends are not adequate in many cases. For example, it is not indicated in Fig 4, which doses of MDM2 and IR were used? The method for measurement of binding of MDM2i to p53 peptide is not described in Figure legend, nor in Results section. No controls have been provided for this experiment either.

We have now improved the figure legends and added the requested information. Note that we have used the same radiation dose across all our *in vivo* experiments (10Gy). We therefore added this information to the Method section (*Radiation treatment*) instead of rewriting this in every figure legend. The one exception was in the experiment done in cell culture (Figure S7). We now mention the dose of 5Gy in the legend of this figure. In addition, we have now included detailed description of the method for measuring the binding of MDM2i to p53 peptide in the Method section and the use of reference compounds as controls (*MDM2 affinity measurements by Time Resolved Fluorescence Energy Transfer (TR-FRET) Assay*).

References #7 and 21 are the same, as well as ref #8 and 23

We thank the reviewer for catching these duplications, which are now fixed.

Reviewer #3: *In “p53 dynamics vary between tissues and are linked with radiation sensitivity” Ornstein et al examined the effects of total body irradiation on various tissue types and the p53 response in those tissues. The authors examined the p53 pathway over a 7-hour time course in the thymus, spleen, and large and small intestines. In the small intestine, they broke down the expression of various pathway members by cell position as well. They further showed that a novel MDM2 inhibitor was able to induce p53 and reduce xenograft tumor growth relative to vehicles. However, a more expansive analysis of these pathways by cell position both before and after MDM2i treatment is needed.*

We thank the author for their interest in our work. We have added substantial data to further validate the impact of p53 dynamics on the response to radiation across tissues. Specifically, we added new staining data of effectors and regulators of the DNA damage response including γ H2AX, phospho-p53, Ki67, p21 and cleaved caspase 3 (**New Figures 1c-g, 2c, 4d-e, S2 and S3**).

Our analysis revealed systematic differences of the p53 response between tissues and emphasizes the difference between lymphoid tissues and the intestines.

Major Criticism:

Methods:

1. Please state whether the cell position includes Lgr5+ CBCs in the small intestines. If not, it is necessary to include Lgr5+ CBCs.

We attempted 3 different antibodies to detect Lgr5 in the small intestine, but unfortunately none of these work. Instead, we added new staining for another stem cell marker, OLFM4, and showed that it co-localizes to the small intestine crypts together with p53 (**New Supplementary Figure 1**). In addition, we added new staining with the proliferative marker ki67 (**new Figure 1e, f**) showing that replicative cells in the crypt are capable of activating p53.

2. Please include unirradiated controls for TUNEL stained slides in Figure 1.

We now include unirradiated controls for the TUNEL staining in Figure 1 (**New Figure 1h**) and quantification of cells distributions before and after irradiation (**New Supplementary Figure 3**)

Figures:

1. Figure 1C: Please show staining for cleaved caspase-3 in addition to TUNEL to elucidate p53-dependent apoptosis vs. cell death in general.

We added staining for cleaved caspase-3 (**new Supplementary Figure 2**). While the signal was dim, it showed similar trends as the TUNEL staining, strengthening the evidence for p53-dependent apoptosis. In addition, we added citations noting that radiation induced death in tissues primarily result from apoptosis.

2. Figure 2: Please also stain tissues for phospho-p53, p21, MDM2, PUMA, cleaved-caspase 3, TUNEL and BrdU. Please quantify these in the manner of figures 2B, 2C, and 2E.

We attempted staining with most of the antibodies suggested by the reviewers. We successfully stained for phospho-p53, p21 and cleaved-caspase 3. The new images and analysis are now presented in **new Figures 2c, 4d, 4e and Supplementary Figure 2**. We unfortunately were not able to obtain specific detectable signals with MDM2 or PUMA antibodies despite trying 2 antibodies for MDM2 and 3 for PUMA. We did not stain for BrdU, as it alters the radiation response of tissues (e.g. Hou DM et al. 2010 PNAS 107: 18475). Instead, we used Ki67 to detect proliferation state and its impact on p53 activation (**new Figure 1e, f**). We thank the reviewer for this suggestion as these new measurements added to the richness of our work and supported our main findings.

3. Figure 2: Please clarify whether, for small intestine cell positions, $n > 100$ cells is total or per position.

The number of cells was >100 per time-point. We now included this in the figure legend.

4. Figure 2: Please repeat the experiments in this Figure using p53^{-/-} mice.

The essential role of p53 in response to radiation has been extensively studied and is well established. Here, we focused on a specific mechanism linking p53 with the response to

radiation; the dynamics of wild-type p53, which cannot be investigated in p53 $-/-$ mice. We added references to strengthen the known role of p53 in response to radiation *in-vivo*.

5. Figure 4: Please repeat the staining of the small intestines suggested in Figure 2 in mice treated with IR +/- Mdm2i to elucidate mechanisms and effects of MDM2i on normal tissue response to DNA damage.

We now include new quantitative analysis linking MDM2 inhibition to increased expression of p53, phospho-p53, and p21 in the small intestine (**new Figure 4c-e**). These results show that MDM2i is effective in enhancing the p53 response in normal tissues.

Discussion:

1. Please address what makes this novel MDM2 inhibitor different from others in use. Does it have unique biochemical properties that allow for it to be administered as a single dose, or is that simply an untested method?

The inhibitor we used is highly effective, but the major difference here is the direct combination with radiation, which was not tried in a single dose format previously. We now better clarify this point in the text. In addition, we now show the efficacy of this inhibitor in a second mouse model (**new Supplementary Figure 5**).

Minor Criticism:

1. In all of your figures of stained tissues, please avoid red and green coloring.

We changed the color scheme to a standard blue for DAPI, with either green or red for the protein of interest. For co-staining experiments (**New Figure 1 c, e**) we had to use all 3 colors and included quantifications of the images (**New Figure d, f**).

REVIEWERS' COMMENTS

Reviewer #1 (Remarks to the Author):

The revised version of this paper addresses some of my original technical comments satisfactorily. However, the main conclusion and potential novelty of the paper is still not supported by the data. The authors still claim that differences in p53 dynamics account for the different responses to radiation of various organs. But they still cannot link the two. There are no experiments that can conclusively establish that specifically affecting the dynamics of the TF results in different responses by the tissues or organs.

They recognise that this is a difficult thing to do experimentally, and they mention they are working on transgenic approaches for future studies. This is fine, but it still leaves the main point open. A more conclusive experiment could have been to manipulate p53 dynamics (for example by performing mutations previously known to affect the dynamics, such as mutations of binding sites, etc), and then show that this affects organ radiation sensitivity.

Again, I recognise that it is difficult for everyone in this field to directly test and prove that specific changes in p53 dynamics affect cell behaviours, especially in vivo. But until new approaches are established, we still need to wait before claiming that changes in TF dynamics affect organ behaviours in vivo. As such, I think that the main conclusion (stated in their 'one sentence summary') is not experimentally tested.

Reviewer #2 (Remarks to the Author):

In the revised version of the manuscript the authors added several important details and answered some of my comments. However, the study still failed to establish the link between p53 oscillation, p53-dependent tissue-specific gene expression and cell fate. How different p53 dynamics affects gene expression? What are the key genes which are affected by different p53 dynamics which are important for the cell fate? p21 is NOT the answer to all these questions, especially in light of the recent paper, showing that p21 is not the universal p53 target gene induced in all tissues in vivo

(Moyer SM, Wasylshen AR, Qi Y, Fowlkes N, Su X, Lozano G. Proc Natl Acad Sci U S A. 2020 Sep 22;117(38):23663-23673).

Single-cell RNA-seq should be performed in order to get at least some understanding of how p53 dynamics affects cell fate. Tissue-specificity is explained by different epigenetic state of the cells and by the presence of tissue-specific transcriptional cofactors of p53 – how does dynamics fit in this?

Reviewer #3 (Remarks to the Author):

The manuscript is indeed improved after the revision. However, the data in the revised manuscript lead to several major questions that I hope the authors would be willing to address.

1. Intestinal crypts that expressed high p53 showed less H2Ax after irradiation. What are the underlying mechanisms? Examination of DNA repair pathways, for example, would be helpful.
2. It has been shown that an Mdm2 inhibitor protects the small intestines from radiation. Would the Mdm2 inhibitor that the authors developed also have the same radioprotection effect on the small intestines. If yes, what are the underlying mechanisms?

Reviewer #1

The revised version of this paper addresses some of my original technical comments satisfactorily.

We are pleased the reviewer found our additional data and explanations effective.

...the main conclusion and potential novelty of the paper is still not supported by the data. The authors still claim that differences in p53 dynamics account for the different responses to radiation of various organs. But they still cannot link the two. There are no experiments that can conclusively establish that specifically affecting the dynamics of the TF results in different responses by the tissues or organs.

We thank the reviewer for their direct comment and think that their concern goes to the heart of the paper and the state of the field. We agree that many of our results here are correlative, but we suggest that they amount to a significant step forward for the field. We show that different tissues *in vivo* show dramatically different dynamical patterns of p53 levels and activity. This novel finding is critical for understanding dynamics of signaling in the body. Our data underlines the complexity of p53 signaling in the body and shows that combinations of small molecules and radiation can manipulate those dynamics.

*They recognise that this is a difficult thing to do experimentally, and they mention they are working on transgenic approaches for future studies. This is fine, but it still leaves the main point open. A more conclusive experiment could have been to manipulate p53 dynamics (for example by performing mutations previously known to affect the dynamics, such as mutations of binding sites, etc), and then show that this affects organ radiation sensitivity. Again, I recognise that it is difficult for everyone in this field to directly test and prove that specific changes in p53 dynamics affect cell behaviours, especially *in vivo*. But until new approaches are established, we still need to wait before claiming that changes in TF dynamics affect organ behaviours *in vivo*. As such, I think that the main conclusion (stated in their 'one sentence summary') is not experimentally tested.*

While our findings strongly points to the existence of links between p53 dynamics and tissue fate we agree that our one-sentence summary is too strong and have therefore modified it to: 'The temporal dynamics of p53 *in vivo* are connected with radiation sensitivity and tumor clearance'

Reviewer #2

*In the revised version of the manuscript the authors added several important details and answered some of my comments. However, the study still failed to establish the link between p53 oscillation, p53-dependent tissue-specific gene expression and cell fate. How different p53 dynamics affects gene expression? What are the key genes which are affected by different p53 dynamics which are important for the cell fate? p21 is NOT the answer to all these questions, especially in light of the recent paper, showing that p21 is not the universal p53 target gene induced in all tissues *in vivo* (Moyer SM, Wasylshen AR, Qi Y, Fowlkes N, Su X, Lozano G. Proc Natl Acad Sci U S A. 2020 Sep 22;117(38):23663-23673).*

We thank the reviewer for their acknowledgement of our effort to address their comments. We agree that more work is needed to identify the key genes involved in p53 regulation and function *in vivo* (and further agree that it is not all p21!). We note, however, that this is a perennial problem in the p53 field as even not all the key tumor suppression targets have been identified over the last four decades. Our work suggests how one might start to look for these targets, at what time-points, and in which tissues, and as such is a critical contribution to the literature in this area.

Single-cell RNA-seq should be performed in order to get at least some understanding of how p53 dynamics affects cell fate. Tissue-specificity is explained by different epigenetic state of the cells and by the presence of tissue-specific transcriptional cofactors of p53 – how does dynamics fit in this?

We agree that single cell RNAseq is a promising approach to unlock the link between tissue type and p53 signaling. Unfortunately, time-course single cell RNAseq at high depth would be an immensely complex, expensive, and challenging dataset to assemble, and therefore is beyond the scope of this work. Further, currently there are few good tools to interpret time-course data of this type. Ultimately, we expect that time-course spatial profiling approaches may reveal some of the answers about how dynamics and tissue type interact to generate phenotypes. We included a comment in the discussion of the manuscript about the promise of this approach.

Reviewer #3

The manuscript is indeed improved after the revision. However, the data in the revised manuscript lead to several major questions that I hope the authors would be willing to address.

We thank the reviewer for their interest and support.

1. Intestinal crypts that expressed high p53 showed less H2Ax after irradiation. What are the underlying mechanisms? Examination of DNA repair pathways, for example, would be helpful.

We agree that determining the specific repair pathways activated in the villi compared to the crypts is an interesting research direction. As we discussed in the manuscript, we suspect it relates to the proliferative nature of the cell (for example access to HR repair in cycling cells). Determining the underlying mechanism and origin of heterogeneity within tissues is however beyond the scope of this work, which focuses on comparisons between tissues.

2. It has been shown that an Mdm2 inhibitor protects the small intestines from radiation. Would the Mdm2 inhibitor that the authors developed also have the same radioprotection effect on the small intestines. If yes, what are the underlying mechanisms?

The model of radio protection by MDM2 inhibitors is that pre-treatment with these small molecules results in protection to subsequent radiation as it places the crypt cells in an arrested state. We expect in principle that the NMI801 compound would lead to similar effects as other MDM2 inhibitors if administered pre-treatment. Note that in our study, the MDM2 inhibitor was applied

post-treatment, and since no pre-existing arrest state was induced, it did not provide a protective effect.